# Split-Belt Training but Not Cerebellar Anodal tDCS Improves Stability Control and Reduces Risk of Fall in Patients with Multiple Sclerosis

**DOI:** 10.3390/brainsci12010063

**Published:** 2021-12-31

**Authors:** Carine Nguemeni, Shawn Hiew, Stefanie Kögler, György A. Homola, Jens Volkmann, Daniel Zeller

**Affiliations:** 1Department of Neurology, University Hospital of Würzburg, Josef-Schneider-Str. 11, 97080 Würzburg, Germany; shawnhiew18@gmail.com (S.H.); stefaniekoegler@gmx.de (S.K.); Volkmann_J@ukw.de (J.V.); Zeller_D@ukw.de (D.Z.); 2Department of Neuroradiology, University Hospital of Würzburg, Josef-Schneider-Str. 11, 97080 Würzburg, Germany; Homola_G@ukw.de

**Keywords:** multiple sclerosis, split-belt treadmill, cerebellar tDCS, gait, balance, risk of fall

## Abstract

The objective of this study was to examine the therapeutic potential of multiple sessions of training on a split-belt treadmill (SBT) combined with cerebellar anodal transcranial direct current stimulation (tDCS) on gait and balance in People with Multiple Sclerosis (PwMS). Twenty-two PwMS received six sessions of anodal (PwMS_real_, *n* = 12) or sham (PwMS_sham_, *n* = 10) tDCS to the cerebellum prior to performing the locomotor adaptation task on the SBT. To evaluate the effect of the intervention, functional gait assessment (FGA) scores and distance walked in 2 min (2MWT) were measured at the baseline (T0), day 6 (T5), and at the 4-week follow up (T6). Locomotor performance and changes of motor outcomes were similar in PwMS_real_ and PwMS_sham_ independently from tDCS mode applied to the cerebellum (anodal vs. sham, on FGA, *p* = 0.23; and 2MWT, *p* = 0.49). When the data were pooled across the groups to investigate the effects of multiple sessions of SBT training alone, significant improvement of gait and balance was found on T5 and T6, respectively, relative to baseline (FGA, *p* < 0.001 for both time points). The FGA change at T6 was significantly higher than at T5 (*p* = 0.01) underlining a long-lasting improvement. An improvement of the distance walked during the 2MWT was also observed on T5 and T6 relative to T0 (*p* = 0.002). Multiple sessions of SBT training resulted in a lasting improvement of gait stability and endurance, thus potentially reducing the risk of fall as measured by FGA and 2MWT. Application of cerebellar tDCS during SBT walking had no additional effect on locomotor outcomes.

## 1. Introduction

Multiple Sclerosis (MS) is a chronic neurological disease affecting the central nervous system (CNS). It is characterized by inflammation, demyelination, and consecutive decline in various neurologic functions [1]. Gait dysfunction in MS is distinguished by a decrease in gait speed, walking endurance, and step length, as well as by an increase in variability of gait. It is estimated that 40–60% of PwMS show gait impairment which interferes with daily activities [2]. Consequently, gait and balance impairments lead to an increased risk of falling, with over half of PwMS reporting falling within a three- to six-month period, and of those, 30% reporting multiple falls leading to injuries [3,4]. Therefore, it comes as no surprise that gait impairments and a heightened fear of falling are leading factors contributing to the inability of affected individuals to continue working and to stay active and socially connected, thereby compromising mental health and quality of life for PwMS [5].

Previous work has suggested that despite an impaired overall motor performance, PwMS retain the ability to acquire new motor skills [6,7,8]. More recently, we demonstrated that the motor adaptive capacity in people with mild to moderate MS remains comparable to healthy individuals [9]. Adaptation is the ability to learn via trial-and-error feedback and can be evaluated by inducing a perturbation, for example by changing the speed of one belt during split-belt treadmill (SBT) walking [10,11]. While motor learning capacity decreases with higher regional injury [12], it remains an ability worth capitalizing on in motor rehabilitation. Complementary methods to maximize the benefit of such training, for example through the enhancement of beneficial neuroplasticity, would be of high value to therapists.

Transcranial direct current stimulation (tDCS) is a non-invasive neuromodulatory method that alters the cortical excitability of target brain areas by delivering weak currents via electrodes on the scalp [13]. These electrical currents increase or decrease neuronal firing rates by modulating their resting membrane potentials [13,14]. Its low cost and quick implementation make it not only a popular research tool but also one with potential for therapeutic use.

The cerebellum has been observed to be crucial for adaptive learning [11,15,16,17,18,19,20]. It plays a key role in planning movement through error prediction, in the execution of planned movement, and in the stability of movements by automatizing movement that was initially attentionally demanding, through practice [21,22,23]. As such, the cerebellum presents a good target for modulation in an adaptive learning paradigm. In a study using an SBT paradigm, anodal tDCS (atDCS) to the cerebellum of healthy participants during the adaptation phase (where the belts moved at different speeds) expedited the adaptive process, while cathodal tDCS slowed down the process [10]. In a more recent study, however, a single session of atDCS to the cerebellum after SBT walking in PwMS was unable to improve adaptation or consolidation of the adaptive learning [9]. However, it has been demonstrated that a single session is usually insufficient to produce benefits [24].

In the present study, we aimed to evaluate the therapeutic potential of multiple sessions of SBT training combined with atDCS to the cerebellum with respect to gait and balance in PwMS, further translating to a reduction in risk of falls and increased mobility in PwMS. We consider the combination of motor training and stimulation to be complementary and a highly promising new therapeutic avenue that could pave the way for a new rehabilitative approach. We hypothesize that compared to sham stimulation, anodal stimulation would significantly improve measures of gait and stability relative to baseline, namely the functional gait assessment scores and overground walking speed.

## 2. Methods

### 2.1. Participants

The study conformed to the principles of the declaration of Helsinki and was approved by the local ethics committee of the Medical Faculty at the University of Würzburg (99/18-Sc). All methods were performed in accordance with the relevant guidelines and regulations.

All participants gave their written informed consent before participating in the experiments and were naive to the purpose of the study.

The results presented here include the analysis of the data of 22 participants. Definite MS following the McDonald criteria [25], Expanded Disability Status Scale (EDSS) score between 2 and 6.5 [26], stable condition within the last three months, and ability to perform the locomotor task were prerequisites to participate in the study. Exclusion criteria included pregnancy, history of addiction, history of depression or psychosis within the last year, treatment with corticosteroids within the last 30 days, other CNS diseases than MS, current use of sedatives, and contraindications for tDCS (e.g., history of epileptic seizures, implanted electrical devices). While disease-modifying and symptomatic treatments were not part of exclusion criteria, it was not allowed to start or withdraw such treatments or change their dosage during the course of the study.

### 2.2. Clinical and Motor Assessment

All PwMS underwent a full clinical assessment including the evaluation of the EDSS score [26], the scale for the assessment and rating of ataxia (SARA [27]), Würzburg fatigue inventory in MS (WEIMuS [28]), frontal lobe function with the Frontal Assessment Battery (FAB [29]) and symptoms of depression (Beck Depression Inventory, BDI [30]). The EDSS measures impairment in eight functional systems and is scored on a scale of 0–10 with possible sub scores of 0.5. A higher EDSS score indicates greater disability with scores under 4.5 reflecting the ability to walk without aid and higher scores reflecting an impairment to walking. The SARA, to evaluate cerebellar ataxia, is made up of eight items, scored on a scale from 0–8 with a higher score indicating greater disability. The WEIMuS, which evaluates MS fatigue, consists of 17 items scored on a scale from 0–4 with higher scores indicating more fatigue. The FAB assesses executive function and comprises six items scored on a scale of 0–4, with higher scores indicating better performance. These assessments were administered by the experimenter once at baseline. The BDI is a 21 question multiple-choice questionnaire where the score is summed across the 21 questions and higher scores indicate more depression with a score greater than 10 indicating mild depression, scores greater than 19 indicating moderate to severe depression [31]. The Pittsburgh Sleep Quality Inventory (PSQI [32]), which has 19 items, was also administered to evaluate baseline sleep quality in the month preceding the experiment, where higher scores indicate poorer sleep quality. These two questionnaires were administered at baseline.

Lower limb function using the timed-walk tests (25 feet [33] and 50 m), the 9-hole peg test (9HPT) for dexterity [34], and handedness [35] were assessed. Additionally, initial gait impairments and the risk of falls were evaluated using the timed 2 min walking test (2MWT) [36], where distance walked was the outcome measure, and the timed-up and go (TUG) test [37], where the time taken to transition from sitting to standing and vice versa was measured. The scores on the functional gait assessment (FGA) [38], administered by the experimenter, and the falls efficacy scale (short-FES [39]), reported by patients. Clinical and motor assessments were completed once each on the initial and the last training session and during the two weeks post-training session.

The participants were pseudonymized and randomly assigned to anodal (PwMS_real_) or sham (PwMS_sham_) cerebellar tDCS.

### 2.3. CNS Injury

The functional integrity of the corticospinal tract in PwMS was tested by evaluating the central motor conduction time (CMCT) of the motor evoked potentials (MEP) in the tibialis anterior muscle using transcranial magnetic stimulation (TMS). The procedure corresponded to the clinical routine for the diagnostic of MS. The CMCT was defined as the difference between the cortical MEP latency and the peripheral motor conduction time [40]. The relationship between CMCT and body height was accounted for, as previously done [40], to determine the upper normal limit.

Additionally, we collected the MRI scans of the participants and were able to analyze 15 of them (PwMS_real_, *n* = 7; PwMS_sham_, *n* = 8). Lesions on T2-weighted MR sequences were outlined by a custom-made semi-automated edge finding software tool (Segmentierungstool, Department of Neuroradiology, University Hospital Würzburg) and, where necessary, corrected manually by the investigator blind to the behavioral assessment of the study.

### 2.4. Experimental Design

The testing paradigm consisted of offline cerebellar tDCS followed by locomotor training. Figure 1A,B presents a diagram detailing the entire experimental timeline.

#### 2.4.1. Offline Anodal Cerebellar tDCS

Offline tDCS was delivered through two 25 cm^2^ surface saline-soaked sponge electrodes to the subjects. Both the participants and the experimenter were blinded to the type of stimulation (sham or anodal) using the “study mode” implemented in the tDCS stimulator (DC-Stimulator-Plus, Neuroconn, Germany). Subjects were randomized to receive anodal (2 mA current for 15 min) or sham tDCS over the cerebellar hemisphere ipsilateral to *fast* leg with the anodal electrode applied over the cerebellum 3 cm lateral to the inion, and the cathodal electrode positioned over the ipsilateral buccinator muscle [10]. These parameters (electrode placement, duration, and current intensity) were chosen as they have been shown to successfully modulate adaptation during SBT in a previous study [10]. Cerebellar tDCS was applied prior to the split-belt training since it is hypothesized that anodal tDCS prior to a task primes the system for response and could induce stronger after effects, lasting beyond the stimulation, leading to an improvement in performance [41].

#### 2.4.2. Split Belt Treadmill Paradigm

Subjects participated in a locomotor adaptation task (LAT) on a custom SBT (Woodway, Waukesha, WI, USA) [10,41,42] with belts moving together (tied-*slow*/tied-*fast*) or at different speeds (split-*slow*/*fast*). The fastest walking speed on the treadmill was individually tailored to the maximum speed obtained overground during the initial timed-walk tests. The less affected or the dominant leg was selected to be the side of the fast belt during the split phases. For all testing, subjects wore comfortable walking shoes and a safety harness. Figure 1B shows the experimental paradigm. Briefly, during the **baseline periods**, subjects walked with both belts tied at a slow speed (tied-*slow*) and fast speed (tied-*fast*) for 2 min each. During the **adaptation period**, they performed asymmetrical training with each leg on one belt walking for 20 min at a speed ratio *slow/fast* of 1:2. After 10 min of adaptation, the belts returned briefly (1 min) to the tied configuration (“**catch trial**”). Following this, the belts were split for another 5 min followed by a second catch trial and the end of the adaptation period (5 min) to complete the total 20 min of adaptation. The catch trials were presented in a random order (tied-*slow* or tied-*fast*) to induce surprise and instability, mimicking obstacles or potential fall inducers. During SBT walking, participants were allowed to hold onto the handrails but were instructed not to lean on the handrails if possible. In the **overground post-adaptation period**, all subjects performed a 50 m walking test to evaluate immediate treadmill training after-effects. Subjects were transported in a wheelchair between the treadmill and overground walking to ensure that no walking other than that planned for experimental purposes would be collected by the sensors between the treadmill and overground periods. Participants that used a walking assisting device to ambulate in their daily life, were allowed to use the same walking aid during the test overground.

The protocol described above (T0) was repeated over six sessions with three sessions per week for two weeks. Each of the three sessions was separated by 48 h (Figure 1A).

#### 2.4.3. Gait Capture and Data Collection

The evaluation of changes in the risk of falls was done overground at T5 using the FGA, the 25 feet walking test (25FWT), the 50 m walking test (50mWT), the 2MWT, and the TUG. Additionally, the long-term effects of the physical therapy were assessed during a final gait evaluation covering the same tests, two weeks after the last training (T6).

All the overground gait assessments were done using the G-Walk 2 system of gait analysis (BTS Bioengineering, Milanese, MI, Italy). This is a portable wireless, inertial system with wearable sensors which is attached to the waistline of the patient during gait assessment. Acceleration data are sampled at 100 Hz frequency, transmitted by Bluetooth to a notebook, and processed using the software program BTS G-Studio (Bioengineering, Milanese, MI, Italy). This system has been used to measure gait parameters in healthy older adults [43], patients with cerebellar ataxia [44], and Parkinson’s disease patients [45].

### 2.5. Data Analysis

The main outcome variable was the change on the FGA which assessed the ability of people to maintain their balance while walking in presence of external demands based on 10 items. We evaluated the change induced by the combinational therapy (tDCS + SBT) in FGA by assessing the mean difference between the scores (∆FGA) at T5 and T6 compared to the baseline between PwMS_real_ and PwMS_sham._ We carried out the same analysis within-subjects after merging the groups, in order to evaluate the effects of treadmill training alone.

The evaluation of the secondary outcome variables, namely TUG, 50mWT, and 2MWT was done following the same procedure assessing between group differences and subsequently intra-subject changes at T5 and T6 for scores, time (s), and distance (m) respectively.

The score of the functional analysis of the risk of falls using the TUG test was calculated automatically by the motion analysis software G-Studio (BTS Bioengineering, Milanese, MI, Italy).

The relationship between the baseline characteristics of the participants (EDSS score, CNS injury) and changes in motor outcomes was assessed using correlation analysis.

### 2.6. Statistics

Data were tested for normality and homogeneity using Shapiro-Wilk and Levene tests respectively and non-parametric statistics were used when data was found to have violated the assumptions for parametric tests.

To test the effects of the combinational therapy, mixed model analysis of variance (ANOVA) with post hoc comparisons was used to assess the differences in risk of fall and motor outcomes between PwMS_real_ and PwMS_sham_ with the stimulation condition as the between-subjects factor. Each stimulation has three levels which are the sessions (T0, T5, and T6), defined as the within-subjects factors.

Two-tailed Spearman’s correlations were used to assess relationships between baseline characteristics and changes in motor outcomes.

To evaluate the independent effect of six sessions of training on the SBT on gait and the risk of falls in PwMS, repeated-measures ANOVA, with sessions (T0, T5, and T6) as the within-subjects factor, and with post hoc comparisons (for T0, T5, and T6) as well as two-tailed *t*-tests (for T5 and T6) were performed with pooled data (PwMS_real_ and PwMS_sham_).

Sphericity was tested using the Mauchly test and a Greenhouse-Geisser correction was applied to adjust for the lack of sphericity where applicable. Bonferroni correction was applied for multiple comparisons. The level of statistical significance for all measures was set at *p* < 0.05, and all statistical calculations were completed using SPSS version 27.0 (IBM, Tokyo, Japan) software.

## 3. Results

### 3.1. Demographics and Clinical Characteristics

Details about the clinical and demographic features of the patients are given in Appendix A and Table 1. Twelve participants received the anodal stimulation (PwMS_real_, 5 females) and 10 patients received the sham stimulation (PwMS_sham_, 7 females). Details about the recruitment process are summarized in the Appendix A.

There was no difference between PwMS_real_ and PwMS_sham_ with regards to age [t(20) = 0.69, *p* = 0.49], disease duration [t(20) = 1.81, *p* = 0.08], 25FWT [t(20) = 1.87, *p* = 0.07] and EDSS score [U = 45.5, *p* = 0.35]. The participants spent the same time performing the 9HPT (9HPT-right hand, *p* = 0.60 and 9HPT-left hand, *p* = 0.57). There was no sign of dysexecutive type dementia and no FAB difference between groups (*p* = 0.32). PwMS_real_ had a significantly higher score on the SARA scale compared to PwMS_sham_ (*p =* 0.04) but the median score in both groups was below 5.5 which is the cut-off score corresponding to mild dependence in the performance of daily living activities [42]. Similarly, PwMS_real_ initially had a significantly higher score on the short-FES scale compared to PwMS_sham_ (*p* = 0.01). Participants in the PwMS_real_ expressed high concern about falling (median score 15 [9–27]) while PwMS_sham_ expressed moderate concern about falling (median score 12 [7–18]) [39,46]. The score on this subjective evaluation decreased and was no longer different between the groups at T6 (PwMS_real_: median score 11 [9–21]; PwMS_sham_: median score 9 [7–18]; *U* = 16, *p* = 0.16).

Based on the answer to the appropriate questionnaires, participants expressed a moderate quality of sleep, significant fatigue, and showed no signs of depression (no group differences with *p* = 0.81, *p* = 0.86, and *p* = 0.61, respectively).

Baseline FGA score negatively correlated with the short-FES (r_s_ = −0.47, *p* = 0.028), but not with sleep quality (r_s_ = 0.30, *p* = 0.18) as expressed by the Spearman correlation. We found no correlation between the baseline scores on the short-FES (TUG, *p* = 0.54; 50mWT, *p* = 0.26; 2MWT, *p* = 0.47), the PSQI (TUG, *p* = 0.06; 50mWT, *p* = 0.91; 2MWT, *p* = 0.17) and the additional three indexes of motor performance evaluated in this study (Table 2).

### 3.2. CNS Injury

Evaluation of CNS injury for PwMS_real_ and PwMS_sham_ is summarized in Table 1.

Six patients in PwMS_sham_ and eight patients in PwMS_real_ presented pathological CMCT for the lower limb. There was no significant difference on the left leg (*t*(16) = −0.10, *p* = 0.92) and the right leg (*t*(16) = −1.18, *p* = 0.27) denoting a similar level of pyramidal demyelination for the individuals with prolonged CMCT.

The analysis of available MRI data showed no difference in the volumes of the ventricular, non-ventricular and the total lesions between PwMS_real_ and PwMS_sham_ (*t*(13) = 1.03, *p* = 0.32; *t*(13) = 2.06, *p* = 0.06 and *t*(13) = 1.24, *p* = 0.24, respectively, Table 2).

### 3.3. Effects of atDCS to the Cerebellum on Gait Stability

The participants received anodal or sham tDCS to the cerebellum followed by training on the SBT as described in Figure 1. The stimulation was well-tolerated without discomfort and no participant reported adverse effects following the session. All the participants were able to complete the task on the SBT.

There was an improvement on the FGA and the TUG, relative to baseline. The effect of the mixed-model ANOVA for changes in the FGA score relative to baseline with the between-subject factor *group (PwMS_real_* vs. *PwMS_sham_)* and the within-subject factor *day (T5*, *T6)* showed no significant *group* × *day* interaction indicating similar changes in the score as a function of day between the groups (*p* = 0.13, Figure 2A and Table 3). We found a significant main effect of *day* underlining that time has a significant effect on the FGA scores in both groups (*p* = 0.02, Figure 2A and Table 3).

Comparison of the outcome on the secondary measures TUG, 2MWT, and 50mWT between the two groups showed no significant *group* × *day* interaction and no significant main effect of *group* (*p* > 0.05 in all parameters, Figure 2B–D and Table 3).

In summary, PwMS_real_ and PwMS_sham_ showed the same level of performance and changes on the motor outcomes independent from tDCS stimulation mode applied to the cerebellum (anodal vs. sham). The combinational therapy comprising the active anodal pre-stimulation with tDCS followed by the training on the SBT did not modulate the risk of falls and the stability control of the participants. However, we observed a significant main effect of *day* on the FGA score and 2MWT in all the participants suggesting a possible effect of the repetition of the training on gait parameters. In order to confirm this result, we pooled data of the participants to evaluate the effect of the repetition of SBT alone.

### 3.4. Repetition of Training on the SBT Reduces Risk of Fall and Improves Endurance in PwMS

Repeated measures ANOVA were used to compare average scores and performance on the various tests at T0, T5, and T6. The analysis of variance showed a significant difference between FGA scores at T0, T5, and T6 (*p* < 0.001, Table 4). Post hoc analysis indicated that the FGA scores at T5 and T6 were significantly higher than the initial baseline score at T0 (*p* = 0.001 and *p* < 0.001, respectively). The difference of FGA scores compared to baseline indicated an improvement of 3.3 and 5.1 points at T5 and T6, respectively, which at T6 exceeds the 4-point threshold for a minimal clinically important difference [47]. Moreover, we found a significant difference between T5 and T6. This result denotes a further improvement after the training has ceased (*p* = 0.010, Figure 3A and Table 4). Similarly, there was a significant improvement in the distance walked within 2 min (2MWT) at T5 and T6. The walked distance improved by 12.31 m and 11.56 m relative to baseline at T5 and T6 respectively. Post hoc comparison indicated a significant difference between T0 and both T5 and T6 (*p* = 0.011 and *p* = 0.035, respectively, Figure 3B and Table 4). The difference between T5 and T6 was not significant (*p* = 0.80).

The analysis of the time spent performing the TUG and the 50mWT did not show any significant change from T0 to T5 or T6 (*p* = 0.28 and *p* = 0.43, respectively, Figure 3C,D, Table 4). There was no significant difference between T5 and T6 (*p* = 0.85 and *p* = 0.65, respectively, Table 4).

## 4. Discussion

The advancement of efficacious therapies for gait impairment and imbalance is of high priority for people with MS (PwMS). We evaluated the therapeutic potential of combining multiple sessions of SBT walking with tDCS to the cerebellum on improving gait stability and reducing the risk of falling in PwMS. Over the course of sessions, changes of locomotor outcome parameters were not different between PwMS receiving cerebellar anodal tDCS and those receiving sham stimulation. However, a significant and lasting improvement of gait was demonstrated for multiple sessions of SBT training, irrespective of the tDCS mode. Thus, while our results clearly indicate beneficial effects of repeated SBT training, they do not show an additional effect of pre-stimulation by cerebellar anodal tDCS.

### 4.1. No Effects of Cerebellar Anodal tDCS on Stability Control and Risk of Falling in PwMS

We found the same level of performance and changes of locomotor outcomes in PwMS independently from the tDCS mode applied to the cerebellum (anodal vs. sham). This result underlines that a combinational therapy comprising an active pre-stimulation with atDCS followed by the training on the SBT did not modulate the gait stability of our participants. This lack of effect could have several reasons.

The first reason may be the polarity of stimulation that we chose for the experiment. In the context of our combinational approach, we applied anodal, rather than cathodal tDCS to the cerebellum. A previous study showed success in expediting the adaptation process on the SBT by applying atDCS to the cerebellum [10]. However, in the present study, we did not evaluate the effect of tDCS on the rate of adaptation on the SBT as that was not the aim of this study. We focused on the assessment of gait improvement overground as this represents the most important goal to achieve during rehabilitation. It is possible that, while atDCS expedites the adaptation process, it may not be beneficial for gait and balance measures obtained after the training session. In a more recent study, cathodal tDCS to the cerebellum was demonstrated to improve balance [48]. At rest, the cerebellum typically exerts an overall inhibitory tone on the primary cortex, in a phenomenon known as cerebellar brain inhibition (CBI) [49], that is reduced after motor learning [10]. CBI is reflected by reduced motor evoked potentials when a transcranial magnetic stimulation (TMS) pulse is delivered to M1 5–7 ms after a pulse was first delivered to the contralateral cerebellum compared to the MEPs evoked by a single pulse to M1 alone [50]. Cathodal stimulation typically results in a reduced CBI which influences motor learning. Therefore, one could hypothesize that cathodal rather than anodal polarity may induce positive changes in gait [48].

A second potential reason is the stimulation site. The cerebellum has been found to be not only a good target site for modulating adaptation [10,49] but also a good intact port of entry when motor regions are compromised, for example in stroke [51]. However, in case cortical motor regions are spared, M1 may be the better target site for improving gait- and balance-related outcomes of training [52]. Novel combinatory approaches, such as priming the system with cerebellar tDCS followed by tDCS to the M1, which has been studied in improving goal-directed movement in patients with cerebellar ataxia [50], has not yet been tested in the context of gait and balance. However, much work needs to be done in order to select effective stimulation durations, polarity, and montages for a complex procedure such as this. As mechanisms such as homeostatic metaplasticity may reverse the direction of intended effects, clever selection of stimulation parameters is crucial in order to achieve the best outcomes [53].

Another point to consider is the timing of the stimulation relative to the behavioral intervention. Stimulation was applied prior to the start of SBT training with the aim of priming the system for the performance of the locomotor adaptation task. However, the timing of the stimulation in relation to a training task has long been debated. The variability of tDCS-related results so far could, at least to some extent, be attributed to the different timings of stimulation used in the different studies, whether before, during, or after a task. It has been postulated that stimulation prior to the performance of a task could prime the system for improved performance and/or learning of a task [54]. On the other hand, tDCS by itself is said to be too weak to induce any behaviorally meaningful changes [55]. If the mechanisms of tDCS are better understood, tDCS protocols can be cleverly planned in order to reap the most benefit. For example, Andani et al. [48] applied tDCS for a total of 20 min, beginning 5 min prior to a stance task, continuing during the task, and ending 5 min after, to investigate the effects of cathodal tDCS to the cerebellum combined with the provision of visual feedback on balance control. They found a reduced overall body sway when cathodal tDCS to the cerebellum was combined with visual feedback. Their protocol, therefore, primed the system for task performance, boosting the ongoing task and enhancing consolidation.

Finally, activity-dependent modulation of synaptic plasticity also known as metaplasticity would need to be controlled in future studies as this property varies from individual to individual and might modify the response to an external stimulation [56]. Indeed, despite optimization of external non-invasive brain stimulation parameters, the effects of the stimulation would still require the brain to be in an internal enhanceable state for plastic changes to happen and for these effects to be behaviorally evaluable.

### 4.2. Multi-Session Split-Belt Treadmill Training in Endurance and Gait Stability

Our main outcome variable was the functional gait assessment (FGA) score which assesses the ability to maintain balance while walking in presence of external demands. PwMS showed an improvement of 3.3 and 5.1 points at T5 and T6, respectively, relative to baseline. The FGA test has been shown to reliably predict falls in MS, Parkinson’s disease, and elderly people [57,58,59]. Additionally, an increase of 4 points on this assessment is considered clinically relevant in older adults [47]. Therefore, the improvement obtained after six sessions of training on the SBT could translate into meaningful gait changes in the daily life of patients. Moreover, the continuing increase from 3.3 to 5.1 points between sessions T5 and T6 may indicate long-lasting sustained effects of this intervention.

The 2MWT showed a significant improvement in the endurance of patients following the training. The distance walked was increased by over 10 m and this gain was sustained two weeks after the cessation of the training (T6). Gijbels and colleagues have proposed the 25FWT and the 2MWT as standards for evaluating walking capacity in mild-to-moderate PwMS (EDSS 0–6.5), as those tests show a high correlation and an estimation error below 5% in both subgroups [60]. A multicenter study reported the minimal important change following rehabilitation of PwMS as an increase of 9.6 m and 6.8 m from the patient’s and therapist’s perspective, respectively [61]. These data confirm the significance of the results obtained with the SBT in our patients. Moreover, since the training improves endurance, reflected by improvements in the 2MWT, targeting a site known to modulate fatigue could further boost the outcome of the training on endurance.

Identifying the most sensitive tests for regular clinical use and optimal rehabilitation of PwMS remains a challenge. The gold standard for measuring walking mobility is the habitual walking performance (HWP), where the patient’s day-to-day steps are measured using an accelerometer in their own living environment [62]. Distance walked during long walking tests such as the 6 min walking test (6MWT) and the 2MWT have been found to be highly predictive of HWP in PwMS [63]. Hence, the positive effects of our SBT training on the 2MWT suggest that SBT training could improve HWP and therefore improve endurance and fatigue in mild-to-moderate PwMS.

In summary, SBT training could improve balance and endurance during walking, reduce the risk of falling, consequently leading to reduced fear of falling and increased engagement in social and physical activity in PwMS. Since SBTs are fairly cost-efficient, relative to other neurorehabilitative devices, and can be easily and safely implemented without much training, SBT training paradigms show promise in being more commonly integrated into the rehabilitation of PwMS with gait and balance impairments.

### 4.3. Limitations and Future Directions

The present study included a mix of subjects with mild (EDSS ≤ 4; *n* = 13) to moderate (EDSS ≥ 4.5; *n* = 9) MS as well as a mix of MS subtypes. Greater atrophy in the cerebellum has been demonstrated in the progressive subtypes as compared to the relapsing-remitting subtype [64]. For this reason, it remains possible that interventions such as tDCS and SBT may be more effective in particular subtypes or stages of the disease. Due to a fairly small sample, we were unable to differentially analyze the effects of tDCS and of the SBT training in sub-groups (by EDSS score or by disease sub-type). Future studies should investigate the effects of interventions such as tDCS or SBT in different subtypes or at different stages of MS. Additionally, regional grey matter volumes, in particular of the cerebellum and motor regions such as the M1, should be included in future investigations. These steps would allow us to identify patients who would benefit most from SBT training and to define when such an intervention would be most beneficial. While the FGA has been shown to be a good predictor of fall risk, additional direct measures would be necessary to draw any conclusions regarding fall risk. These are for example the center of pressure (CoP) which has been shown to be a good predictor of fall risk. To truly evaluate the lasting effects of the training, several additional follow-ups at even later time points would prove beneficial. Finally, a closer look into more effective tDCS parameters, for example, the current density, are warranted. Several studies have indicated an effect of tDCS when higher current intensities are applied (4 mA vs. 2 mA) [65].

## 5. Conclusions

Multiple sessions of SBT training can improve gait stability and endurance in PwMS, leading to improved balance and a reduced risk of falling. The inclusion of SBT training should be considered in the therapy of PwMS. Systematic investigations of different tDCS parameters are truly needed to identify parameters that will be able to further boost the effects of SBT training. Further, tailored rehabilitative approaches should not just target sites directly involved in the training program but also consider improving other symptoms that could affect the rehabilitative session such as fatigue and pain.

## Figures and Tables

**Figure 1 brainsci-12-00063-f001:**
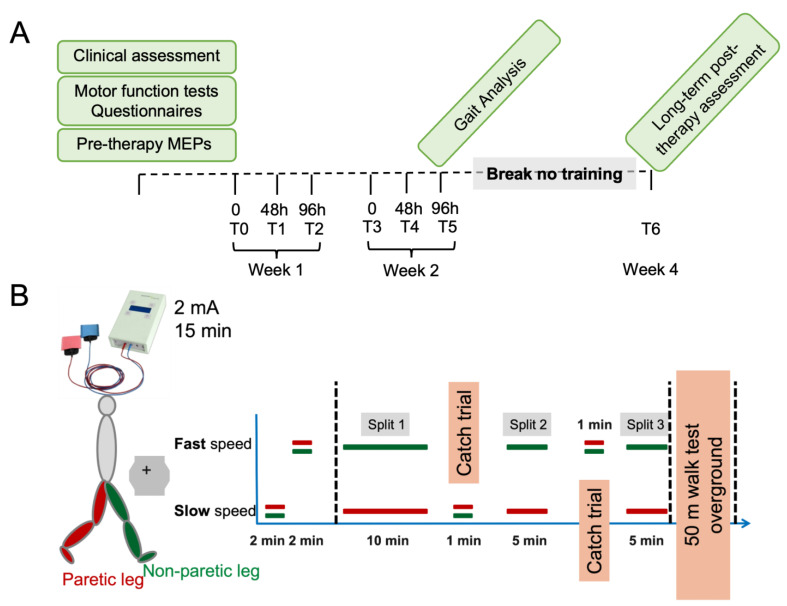
Experimental design. (**A**) Baseline clinical assessments, motor function tests, questionnaires, and motor evoked potentials (MEP) were acquired prior to the start of the training. The experiment consisted of six sessions over two weeks, with three sessions each week separated by 48 h. Gait measures were collected at baseline (T0), and the last training session (T5). After the last training session, there was a two-week break before the final follow-up (T6). The study lasted four weeks. (**B**) For the split-belt treadmill walking, the *slow* leg was determined as the participants’ more-affected or non-dominant leg. The fast speed was determined as the participants’ fastest speed on the 50 m overground walking test, while the slow speed was a third this speed. Participants walked with the belts tied at the slow speed for 2 min and then at the fast speed for a further 2 min. This was followed by the first split trial, lasting 10 min, a slow catch trial, where the belts were tied, for 1 min, and two subsequent split trials lasting 5 min each separated by a fast catch trial for 1 min. Immediately after, participants performed a 50 m overground walking test.

**Figure 2 brainsci-12-00063-f002:**
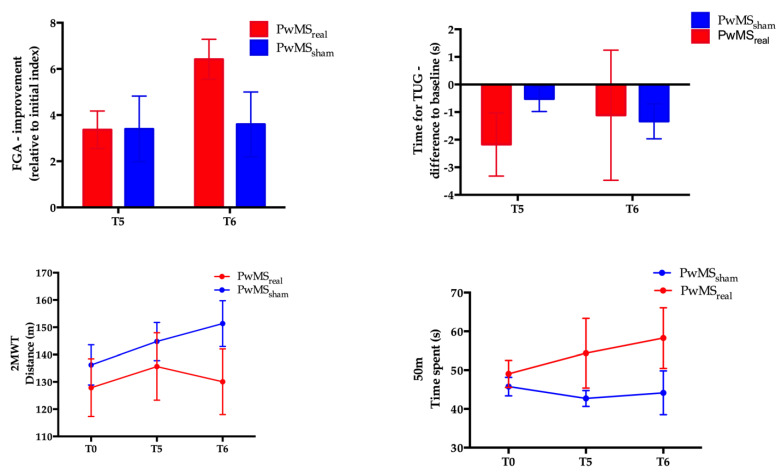
(**A**) Change in functional gait assessment (FGA) scores at T5 and T6 relative to baseline for PwMS_real_ and PwMS_sham_. (**B**) Change in time for the timed-up and go (TUG) at T5 and T6 relative to baseline for PwMS_real_ and PwMS_sham_. (**C**) Distance walked in 2 min at T0, T5, and T6 for PwMS_real_ and PwMS_sham_. (**D**) Time taken to complete the 50 m overground walking test at T0, T5, and T6 for PwMS_real_ and PwMS_sham_. Note: Error bars represent the standard error of the mean.

**Figure 3 brainsci-12-00063-f003:**
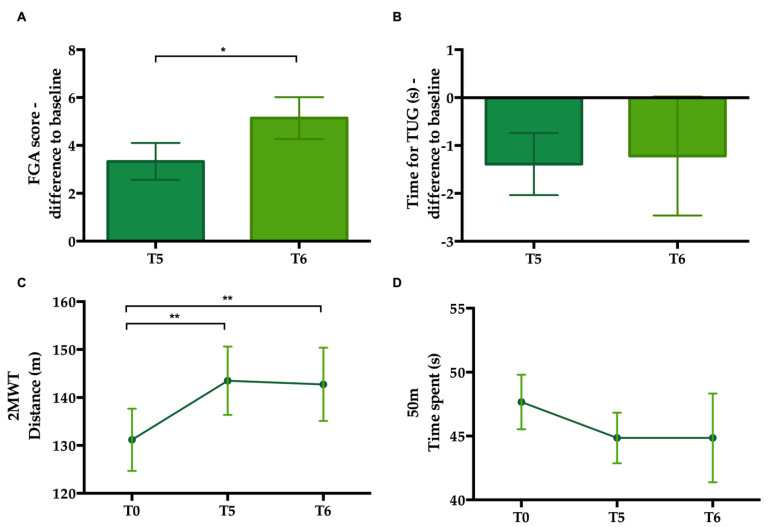
Gait measures for all PwMS pooled. (**A**) Change in functional gait assessment (FGA) scores at T5 and T6 relative to baseline. (**B**) Change in time for the timed-up and go (TUG) at T5 and T6 relative to baseline. (**C**) Distance walked in 2 min at T0, T5, and T6. (**D**) Time taken to complete the 50 m overground walking test at T0, T5, and T6. Note. Error bars represent the standard error of the mean. * *p* < 0.05; ** *p* < 0.01.

**Table 1 brainsci-12-00063-t001:** Demographic, clinical, neurophysiological, and functional data are given for each group (PwMS_real_ and PwMS_sham_). Continuous data are expressed as mean ± standard deviation and ordinal data as median [range].

Clinical Parameters	PwMS_real_ (*n* = 12)	PwMS_sham_ (*n* = 10)	Statistics
**Demographic variable**			
Age [a]	49.83 ± 10.46	46.90 ± 9.00	*p* = 0.49
Gender	7 M, 5 F	3 M, 7 F	
**Clinical neurologic and motor assessment**			
25FWT (s)	8.79 ± 3.90	6.19 ± 1.70	*p* = 0.63
EDSS	4.0 [3–6.5]	3.5 [2–5]	*p* = 0.35
9HPT—right hand (s)	30.60 ± 14.67	27.49 ± 12.74	*p* = 0.60
9HPT—left hand (s)	29.70 ± 7.70	33.81 ± 23.38	*p* = 0.57
FAB	17.0 [16–18]	16.5 [13–18]	*p* = 0.32
SARA	4.0 [1–17.0]	2.0 [1–8]	*p* = 0.04 *
FES	15.0 [9–27]	12.0 [7–18]	*p* = 0.01 *
**Questionnaires**			
PSQI	7 [4–15]	7 [2–15]	*p* = 0.81
WEIMuS	31.5 [5–65]	34.5 [8–54]	*p* = 0.86
BDI	10 [4–26.0]	8 [4–22.0]	*p* = 0.61
**CNS lesions ^#^**			
ΔCMCT right leg (ms)	−7.80 ± 9.41 (*n* = 9)	−3.28 ± 6.66 (*n* = 9)	*p* = 0.27
ΔCMCT left leg (ms)	−8.99 ± 8.42 (*n* = 9)	−8.53 ± 11.30 (*n* = 9)	*p* = 0.92
Ventricular lesion volume (mm^3^)	8997.99 ± 11157.86 (*n* = 7)	4147.83 ± 6866.08 (*n* = 8)	*p* = 0.32
Non-ventr. lesion volume (mm^3^)	1798.64 ± 952.33 (*n* = 7)	776.94 ± 960.31 (*n* = 8)	*p* = 0.06
Total lesion volume (mm^3^)	10,796.63 ± 11,212.53 (*n* = 7)	4924.77 ± 6866.082 (*n* = 8)	*p* = 0.24

Note: ^#^ CNS lesions: based on the consent of the patient for motor evoked potentials and on evaluable MRI scans. Abbreviations: 25FWT, 25 feet walking test; EDSS, Expanded Disability Status Scale; 9HPT, 9-hole peg test; FAB, Frontal Assessment Battery; SARA, scale for the assessment and rating of ataxia; FES, falls efficacy scale; PSQI, Pittsburgh Sleep Quality Inventory; WEIMuS, Würzburg fatigue inventory in MS; BDI, Beck Depression Inventory; ΔCMCT, difference between individual central motor conduction time and upper normal limit; non-ventr., non-ventricular. * Significant difference with *p* < 0.05.

**Table 2 brainsci-12-00063-t002:** Spearman’s correlation assessing the relationship between the baseline scores on the short Falls Efficacy Scale (short-FES), the Pittsburgh Sleep Quality Index (PSQI), and four indexes of motor performance (FGA score, time on the TUG, distance walked in the 2MWT, and time on the 50mWT) in people with multiple sclerosis.

Variable		Short-FES		PSQI	
	N	r_s_	*p*	r_s_	*p*
**FGA**	22	−0.47 *	0.028	0.30	0.18
**TUG**	22	0.14	0.54	−0.42	0.06
**2MWT**	22	−0.26	0.26	0.03	0.91
**50mWT**	22	0.17	0.47	−0.31	0.17

Abbreviations: 2MWT, timed 2 min walking test; 50mWT, 50 m walking test; FGA, functional gait assessment; N, sample size; TUG, Timed Up and Go. * *p* < 0.05.

**Table 3 brainsci-12-00063-t003:** Two-way mixed-model ANOVA examining the changes from baseline in FGA score (ΔFGA) and time spent performing the TUG (ΔTUG) as well as performance on the 50mWT and 2MWT with the between-subject factor *group* (PwMS_real_ vs. PwMS_sham_) and the within factor *day* (T0, T5, T6).

Measure	Source	Correction	Df	F	*p*	η^2^_p_
**ΔFGA**	**Within-group effects**	GG				
	Day (T5, T6)		1	6.65 *	0.02	0.28
	Day × group		1	2.52	0.13	0.13
	Error day		17			
	**Between-group effects**					
	Group		1	1.56	0.23	0.08
	Error group		17			
**ΔTUG**	**Within-group effects**	GG				
	Day (T5, T6)		1	0.01	0.91	0.00
	Day × group		1	1.07	0.32	0.06
	Error day		17			
	**Between-group effects**					
	Group		1	0.10	0.76	0.01
	Error group		17			
**2MWT**	**Within-group effects**					
	Day (T5, T6)		2	7.90 **	0.002	0.36
	Day × group		2	1.53	0.23	0.10
	Error day		28			
	**Between-group effects**					
	Group		1	0.50	0.49	0.03
	Error group		14			
**50mWT**	**Within-group effects**	GG				
	Day (T5, T6)		1.12	0.69	0.43	0.04
	Day × group		1.12	0.51	0.50	0.03
	Error day		17.92			
	**Between-group effects**					
	Group		1	0.48	0.50	0.03
	Error group		16			

Abbreviations: ΔFGA, change from baseline in functional gait assessment score; ΔTUG, change in timed up and go; 2MWT, timed 2 min walking test; 50mWT, 50 m walking test; Df, degrees of freedom; GG, Greenhouse-Geisser correction; N_PwMSreal_ = 12, N_PwMSsham_ = 10. * *p* < 0.05; ** *p* < 0.01.

**Table 4 brainsci-12-00063-t004:** Repeated measures ANOVA examining the effect of session on FGA scores, time on the TUG and 50mWT, and distance on the 2MWT at T0, T5, and T6 (upper part of the table). Paired-samples *t*-tests examining the changes from baseline in FGA score, time on the TUG and 50mWT, and distance on the 2MWT (difference between T5 and T6; lower part of the table). Pooled data across all PwMS.

Measure	Time	Mean	SEM	Correction	F	*p*	η^2^_p_
FGA	T0	17.95	1.39				
	T5	21.29	1.30		23.29 **	<0.001	0.54
	T6	23.10	0.89				
TUG	T0	11.99	1.87				
	T5	10.67	1.25	GG	1.27	0.28	0.06
	T6	10.77	1.17				
2MWT	T0	131.19	6.49				
	T5	143.50	7.11		7.48 **	0.002	0.33
	T6	142.75	7.63				
50mWT	T0	47.67	2.14				
	T5	44.85	1.98	GG	0.69	0.43	0.04
	T6	45.65	3.46				
**Measure**					**t**	** *p* **	**d**
ΔFGA	T5	3.33	0.77				
	T6	5.14	0.87		−2.87 *	0.010	0.48
ΔTUG	T5	1.39	0.65				
	T6	1.22	1.24		0.19	0.85	0.04
Δ2MWT	T5	12.31	3.59				
	T6	11.56	4.03		0.25	0.81	0.02
Δ50mWT	T5	2.62	0.95				
	T6	3.74	2.86		−0.46	0.65	0.67

Abbreviations: FGA, functional gait assessment; TUG, timed up and go; 2MWT, timed two-minute walking test; 50mWT, 50 m walking test; ΔFGA, change in functional gait assessment; ΔTUG, change in timed up and go; Δ2MWT, change in time on the timed two-minute walking test; Δ50mWT, change in time on the 50 m walking test; ANOVA, Analysis of Variance; GG, Greenhouse-Geisser correction; PwMS, people with multiple sclerosis; SEM, standard error of mean, N = 22; * *p* < 0.05; ** *p* < 0.01.

## Data Availability

The data presented in this study are available on request from the corresponding author.

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
