# Peer review of "Split-Belt Training but Not Cerebellar Anodal tDCS Improves Stability Control and Reduces Risk of Fall in Patients with Multiple Sclerosis"

_brainsci, 2021, doi:10.3390/brainsci12010063_

Round 1

Reviewer 1 Report

While the results and conclusions of your study are well-presented, there are certain  aspects requiring some clarification, i.e. how did you approach patients included in your series who were taken antispasticity agents, or fampridine products. Did not perceive these were part of the exclusion criteria. 

Also, would like to see a comment on the logistics dealing with patients with an EDSS of 6.5 (this level of disability was included in your inclusion criteria) and the utilization of the split-belt treadmill by a person requiring of an assitive device to ambulate.  

Reviewer 2 Report

Summary: The authors present their manuscript which aimed to assess if administering cerebellar tDCS before a split-belt treadmill walking task significantly affected motor function in people with mild-moderate MS. Overall, the study is interesting and will be relevant to the tDCS literature. Nevertheless, I have several comments that I think will improve the impact of the manuscript.

  1. Abstract: Please provide the actual p-values for the results instead of p > 0.05 and p < 0.05. Also, I do not think reporting the significant beyond two places after the decimal is necessary (e.g., p = 0.01 is preferred to p = 0.010).
  2. Introduction, final paragraph: Here you provide the aim of the study. Do you have a hypothesis you might also include?
  3. Methods, I/E criteria:
    • You have the ability to perform the locomotor task as an inclusion criterion. How did you assess this criterion and how is it different from the EDSS criterion?
    • Did you have an expected age range for subject inclusion?
  4. Section 2.2: A better description of each of these tests, including how many trials were completed and what relevant outcomes were collected for the secondary/gait variables (e.g., distance walked in the 2MWT, time to complete the TUG) should be added to the Methods. Other things to include are how many questions/items comprise the EDSS, SARA, WEIMuS, FAB, BDI, and PSQI scales, how the scores are calculated, and directionality/interpretation of the scores (e.g., higher EDSS reflective of more disability; higher BDI = more depression, BDI > 15 minor depression, etc.).
  5. Section 2.2, first paragraph, first sentence:
    • The Kurtzke citation is different than the rest of the manuscript.
    • You should define all acronyms before using them (e.g., SARA, WEIMuS, BDI).
  6. Section 2.2, first paragraph, final sentence: "... to evaluate quality of baseline sleep quality." You can delete "quality of" to improve the readability of this phrase ("to evaluate baseline sleep quality").
  7. Figure 1:
    • My version of the figure has a window of the figure info overlaying part of the text of section B.
    • The catch trials in the figure legend need more description for the figure to stand alone.
  8. Methods, cerebellar tDCS: The time window in which tDCS is administered, before, during, or after a relevant task, likely has an impact on the efficacy of the stimulation. In the Introduction, you justify stimulating the cerebellum by describing positive results of ctDCS administered to healthy subjects DURING the adaptation phase (online). However, you administered ctDCS BEFORE the task (offline) in your study. Can you provide justification for your choice of administering stimulation before the task instead of during the task?
  9. Methods, SBT description: Early in this paragraph, you have a sentence "The less-affected or the dominant leg was made to move faster." The current placement of this sentence does not fit with the sentences around it. Maybe it goes further down when you discuss the speed ratio?
  10. Methods, gait capture paragraph: Please provide more details about the G-Walk 2 sensor/system for readers unfamiliar with this technology. For example,
    • when (during the session) and where is the sensor placed? If during treadmill walking, what relevant data did you collect during this time and how did you use it in your study?
    • Does it accurately collect gait information on a treadmill?
    • At what rate (Hz) does it collect data?
    • What information does the system output?
    • Is this a reliable and valid form of gait assessment (it is, but providing citations confirming this would strengthen the Methods)?
  11. Section 2.5, secondary outcomes: What specific variables were included from these tests? Only time to complete TUG and 50mWT and distance walked in the 2MWT? Did you assess changes in gait characteristics (e.g., speed, stride/step length, steps in turns)?
  12. Section 2.6: It would be great if you could add measures of effects size for both the ANOVA (e.g., partial eta-squared) and pairwise results (e.g., Cohen’s d).
  13. Section 2.6, paragraph 2: Can you please verify which variables, and their levels, were put in as the between- and within-subject factors for the mixed ANOVA?
  14. Section 2.6: You mention two repeated measures ANOVAs, separated by the Spearman correlation description. The difference between these RM-ANOVAs is not readily obvious to me. One is assessing "risk of all as well as motor performance" and the other is assessing the effect "on gait and risk of falls."
    • Also, why did you need the "two-tailed t-tests (for T5 and T6)?" Did the RM-ANOVA post-hoc tests not already provide this information?
  15. Results: Unfortunately, I do not see any tables in the entire manuscript. To my knowledge, Brain Sciences asks authors to insert tables directly into the text body after their first mention (like you have done with the figures).
  16. Section 3.1, final paragraph: I think Spearman correlations are normally indicated with rho (ρ) instead of sigma (σ).
  17. Section 3.3: I recommend changing the title of this section. Consider "Effects of atDCS to the cerebellum"
  18. Results, paragraph 3: Please provide actual p-values instead of the blanket statement of "p > 0.05."
  19. Section 3.4, first paragraph: I like that you indicated the improvement from T0 to T5 and T6. Can you provide more context for this by indicating if these changes are clinically relevant and/or if they exceed minimally clinically important differences? I see that you provide this information in the Discussion, but a brief mention here would be helpful as well.
  20. Results, final paragraph: It seems the beginning of the sentence is accidentally missing.
  21. Section 4.1, paragraph 2: There is another different citation (Jayaram et al., 2012).
  22. Section 4.1, paragraph 2, final sentence: "... cathodal rather anodal..." should be "... cathodal rather than.."
  23. Section 4.1, final paragraph: At least one of these citations would help justify your choice of doing ctDCS before the task, as per my comment above.
  24. Section 4.1, final paragraph: You detail the Andani et al. study, but do not reveal any of their results; at the same time, you imply that this novel before, during, and after stimulation is superior. Can you please summarize their results to provide more context for recommending this timing paradigm or not?
  25. Section 4.3, limitations/future directions: Another consideration that you might make for future directions is increasing the intensity of the stimulation. In this regard, the group out of the University of Iowa (Workman et al. and Rudroff et al. from 2019 to present) has performed several studies on the viability of 4 mA tDCS, including an abstract that hinted at 4 mA being more effective than 2 mA in pwMS (reference below).
    • Workman CD, Kamholz J, and Rudroff T (2020). P174 Transcranial Direct Current Stimulation (tDCS) with 2 mA and 4 mA for the Treatment of a Multiple Sclerosis Symptom Cluster–a Pilot Study. 131, 4. Clinical Neurophysiology, e112.
  26. Section 4.3, sentence 3: This sentence becomes unclear in the middle. I recommend a revision.
  27. Conclusions: the final sentence looks to have a different font.
  28. Final thought: I think your paper has much to offer the tDCS literature. One thing I recommend that you consider is to check your active stimulation group for responders and non-responders (see Wiethoff et al., 2014 for the concept and Workman et al., 2020 for a study that divided subjects into responders and non-responders). Again, I am not mandating that you include this in the present manuscript; I only recommend it as a potentially interesting concept to explore within your data set.
    • Wiethoff S, Hamada M, and Rothwell JC (2014). Variability in Response to Transcranial Direct Current Stimulation of the Motor Cortex. Brain Stimul 7(3), 468-475.
    • Workman CD, Fietsam AC, Uc EY, and Rudroff T (2020). Cerebellar Transcranial Direct Current Stimulation in People with Parkinson's Disease: A Pilot Study. Brain Sci 10(2), IF: 3.332

Reviewer 3 Report

This manuscript is straight-forward, well-written with the data clearly presented and the conclusions fitting. However, there is only single supplementary figure and single table file, thus it does not appear that Tables 1, 2, 3,4 were uploaded or saved into the review manuscript correctly (I can’t find them).  

Section 2.3-The mention of MRI lesion analysis of 15/22 individuals, it would be beneficial at this point to clarify the number of individuals in PwMSreal and PwMssham

 Section 3.2 mentions 14/15 MRIs showing pathological CMCT for the lower limb. This implies that one individual did not. It would be beneficial to clarify this point and perhaps note what group this individual was in  (PwMSreal or PwMssham) and where their lesions were located if present.

Fig. 1- There is no mention of T6 (under the 4 weeks) Please add.

Fig.1.- Perhaps change “Total” to “Total Session #” or something to clarify as the duplication of T0, T1, T2 on adjacent lines caused a pause. There is ample space in Figure 1A, consideration should be given to extend the time line and just have a single row T0,T1, T2,T3, T4, T5, T6 with a “week 1” bracket around T0, T1, T2 and a “week 2” bracket around T3, T4, T5 and “week 4” T6.

Fig. 1 -remove file size description textbox inside the figure.

Page 4 Section 2.4- Please provide citations if the 2mA current for 15min has been established or used in previous works. It would be beneficial either in section 2.4 or in the discussion section where optimal stimulation parameters are talked about to describe why these parameters were selected.

Figure 2A is missing “*” for any significant findings

Page 10 There is a font change in the last 3 sentences.

In the discussion, consider other possible parameters to determine fast/slow treadmill belt speed. The strategy used was a good one, but with spinal cord injury, the walking speed (and thereby the amount of ascending proprioceptive input into the cord) required the participants to achieve a certain walking speed to activate the central pattern generator neurocircuitry to materialize into positive voluntary movements, could MS individuals be similar, in that a certain walking speed must be achieved to improve stability control, and perhaps too many participants failed to reach that speed.  This is akin to “optimal internal stimulation” parameter requirements versus the need to determine optimal external stimulation parameters (intensity and duration of treatment).

There is a single supplementary figure and single table file. It does not appear that Tables 1, 2, 3, 4 were uploaded or saved into the review manuscript correctly (I can’t find them).  

Round 2

Reviewer 2 Report

The authors have addresesd my comments.